# ViTok-v2: Scaling Native Resolution Autoencoders to 5 Billion Parameters

**Philippe Hansen-Estruch** [1]  **Jiahui Chen** [1]  **Vivek Ramanujan** [2]  **Orr Zohar** [3]  **Yan Ping** [4]  **Animesh Sinha** [5]
**Markos Georgopoulos** [5]  **Edgar Schoenfeld** [5]  **Ji Hou** [5]  **Felix Juefei-Xu** [5]  **Sriram Vishwanath** [6]  **Ali Thabet** [5]

## Abstract

Vision Transformer (ViT) autoencoders have emerged as compelling tokenizers for images, offering improved reconstruction over convolutional tokenizers. However, existing ViT tokenizers cannot explore this landscape as performance degrades outside training resolutions, and reliance on adversarial losses prevents stable scaling. ViTok (Hansen-Estruch et al., 2025) found that the compression ratio $r$ mediates a reconstruction–generation trade-off where lower $r$ means better reconstructions but harder generations, so improving tokenizer reconstruction is key to more Pareto-optimal tokenizers. We introduce ViTok-v2, which addresses these limitations with native resolution support via NaFlex for generalization across resolutions and aspect ratios, and a novel DINOv3 perceptual loss that replaces both LPIPS and GAN objectives for stable training at any scale. ViTok-v2 is trained on $\sim$2B images and scaled to 5B parameters, the largest image autoencoder to date. ViTok-v2 matches or exceeds state-of-the-art reconstruction at 256p and outperforms all baselines at 512p and above. In joint scaling experiments with flow matching generators, we show that scaling both the autoencoder and the generator advances the Pareto frontier of this trade-off.

## 1. Introduction

Autoencoders (AEs) (Kingma & Welling, 2013; Oord et al., 2017) compress images into continuous latent representations and serve as a critical component in visual generation (Rombach et al., 2022; Esser et al., 2024). Most production-grade AEs are Convolutional Neural Network (CNN) (LeCun et al., 1989)-based autoencoders, which leverage CNNs' translation invariance to generalize robustly across resolutions and aspect ratios. Recently, ViT (Dosovitskiy et al., 2021) autoencoders have emerged as a compelling alternative, offering more aggressive compression ratios (Yu et al., 2024; Chen et al., 2024), improved reconstruction fidelity (Hansen-Estruch et al., 2025; Sargent et al., 2025; Chen et al., 2025b), and scaling properties (Xiong et al., 2025).

ViTok (Hansen-Estruch et al., 2025) established the compression ratio $r$ as the central axis of the ViT autoencoder design space. This ratio determines the latent dimensionality exposed to downstream generators and mediates a reconstruction–generation trade-off (Ye et al., 2024): lower $r$ improves reconstruction, but produces higher-dimensional latent spaces that are more difficult for diffusion models to learn. Crucially, this trade-off is not fixed. It can be improved along two orthogonal axes: at any fixed $r$, a stronger autoencoder can improve reconstruction without changing the latent space that generators must model, while a larger generator can better exploit the richer latent distributions made available by lower $r$. Therefore, fully exploring this design space requires scaling both autoencoder and generator capacity.

ViTok, however, could only partially explore this landscape. On the autoencoder side, its reliance on adversarial losses introduced training instabilities that made scaling decoders beyond 350M parameters impractical, preventing the study of billion-scale autoencoders. Its fixed 256×256 training regime with absolute positional embeddings led to degradation at other resolutions and aspect ratios, limiting real-world deployment. On the generator side, the reconstruction–generation trade-off was studied with only a single 450M flow model; so it remains unclear how larger generators interact with different compression ratios.

We introduce ViTok-v2, addressing each of these limitations. **(1)** We present, to our knowledge, the **first native aspect-ratio ViT-AE**, integrating NaFlex-style training (Dehghani et al., 2023) with 2D RoPE positional encoding. Unlike prior ViT-AEs that fail outside of their training resolution, ViTok-v2 trained at 256p generalizes to 512p, 1024p. **(2)** We train the **largest continuous ViT image compression**

---

[1]University of Texas, Austin  [2]University of Washington
[3]Stanford University  [4]Spellbrush  [5]Meta Superintelligence Labs
[6]Georgia Institute of Technology. Correspondence to: Philippe Hansen-Estruch <philippehansen@utexas.edu>.

*Proceedings of the 43rd International Conference on Machine Learning*, Seoul, South Korea. PMLR 306, 2026. Copyright 2026 by the author(s).

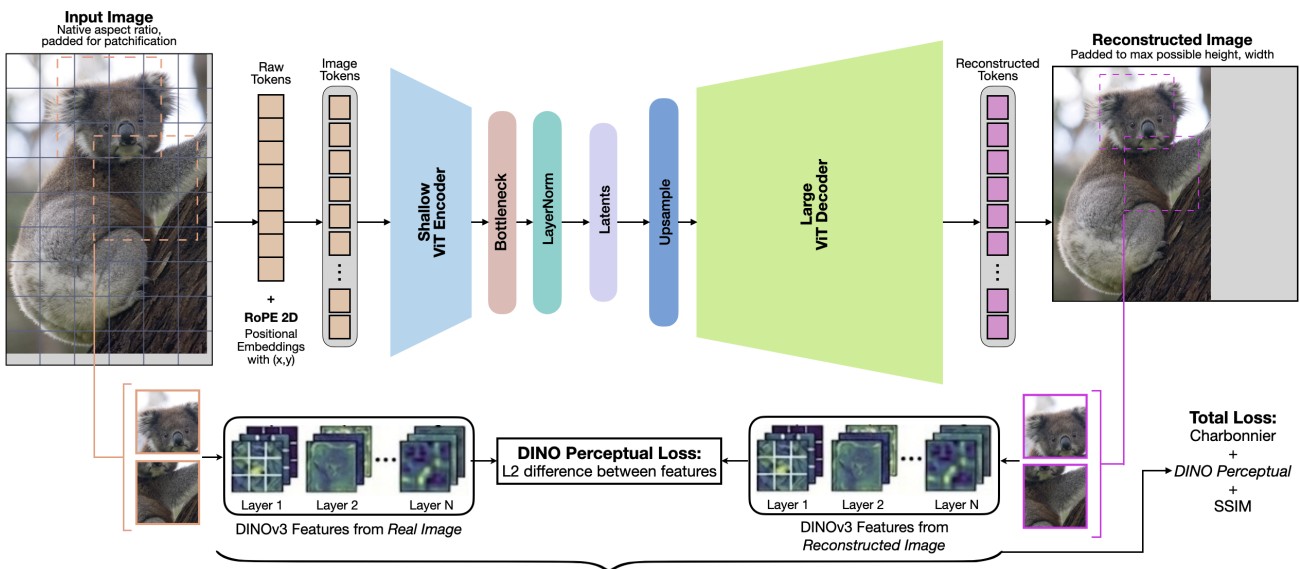

*Figure 1.* **ViTok-v2 method overview.** Input images at native aspect ratios are processed through an asymmetric encoder-decoder architecture: a shallow ViT encoder compresses tokens through a bottleneck into compact latents, which are then upsampled and decoded by a large (5–10× encoder size) ViT decoder. Training combines Charbonnier, SSIM, and DINOv3 perceptual losses on ∼2B images, requiring no GAN or LPIPS losses. **Key advances:** (1) NaFlex training enables generalization to arbitrary resolutions and aspect ratios; (2) models trained with full attention are robust to sliding window attention at inference, enabling efficiency at high resolutions; (3) models scale to 5B parameters, over 14× larger than prior ViT tokenizers.

**autoencoder to date** at 5B parameters (4.5B decoder, 463M encoder), over 14× larger than ViTok's 350M. Our training recipe at this scale enables state-of-the-art reconstruction without adversarial training (Section 3.2). **(3)** We introduce a **novel DINOv3-based perceptual loss** that replaces both LPIPS and GAN objectives. To our knowledge, we are the first to use DINOv3 directly as a perceptual reconstruction loss, enabling stable billion-scale training (Section 3.4). **(4)** We conduct a **joint AE–flow model scaling study**, pairing AEs at 350M and 5B parameters with flow models (Lipman et al., 2023; Esser et al., 2024) at 450M and 1.2B parameters across multiple compression settings, finding that larger generators benefit from lower $r$ while smaller ones perform best at higher $r$; concurrently, Scale-RAE (Tong et al., 2026) demonstrates similar benefits of scaling both components for text-to-image generation (Section 3.6). **(5)** We systematically **compare KL, tanh, and LayerNorm regularization, finding minimal impact on generation quality at scale,** which enables simpler deterministic encoders (Section 3.5).

## 2. Background & Setup

**Autoencoders for latent generation.** An autoencoder consists of an encoder $\mathcal{E}$ that compresses an image $x \in \mathbb{R}^{H \times W \times 3}$ into a latent representation $z = \mathcal{E}(x) \in \mathbb{R}^{h \times w \times c}$, and a decoder $\mathcal{D}$ that reconstructs $\hat{x} = \mathcal{D}(z)$. The bottleneck $z$ has spatial dimensions reduced by factor $f$ with $c$ latent channels per position. The compression ratio $r = 3f^2/c$ measures pixels per latent dimension; lower $r$ means less aggressive compression but makes downstream diffusion

modeling harder. Modern latent diffusion pipelines (Rombach et al., 2022; Esser et al., 2024; Black Forest Labs, 2024) train autoencoders with:

$$\mathcal{L}_{\text{VAE}} = \underbrace{\|x - \hat{x}\|_1}_{\text{recon.}} + \beta \underbrace{D_{\text{KL}}(q(z|x)\|p(z))}_{\text{regularization}} + \lambda_p \underbrace{\mathcal{L}_{\text{LPIPS}}}_{\text{perceptual}} + \lambda_{\text{adv}} \underbrace{\mathcal{L}_{\text{GAN}}}_{\text{adversarial}} \tag{1}$$

The $\beta$-VAE formulation (Kingma & Welling, 2013; Higgins et al., 2017) controls regularization strength, though in practice most models use small $\beta \approx 10^{-6}$.

**Trade-offs in autoencoder training.** Two fundamental trade-offs govern autoencoder design. First, the choice of losses creates tension: L1/L2 reconstruction optimizes pixel-level fidelity (PSNR, SSIM), while perceptual losses (Zhang et al., 2018) and adversarial training (Esser et al., 2021) improve distributional metrics (FID, FDD) at the cost of pixel accuracy, effectively making the decoder partially generative. ViTok (Hansen-Estruch et al., 2025) demonstrated this explicitly: increasing perceptual, adversarial weights improves FID but degrades PSNR/SSIM. Second, a trade-off exists between reconstruction quality (rFID) and generation quality (gFID), mediated by $r$. Decreasing $r$ improves reconstruction but creates a higher-dimensional latent space harder for diffusion models to learn, inducing a parabolic relationship with an optimal $r$ balancing both.

Given these two bottlenecks, our goal is to achieve the best possible reconstruction quality for any fixed compression ratio $r$. We hypothesize that further decoder scaling is the key lever: a larger decoder can better invert the information

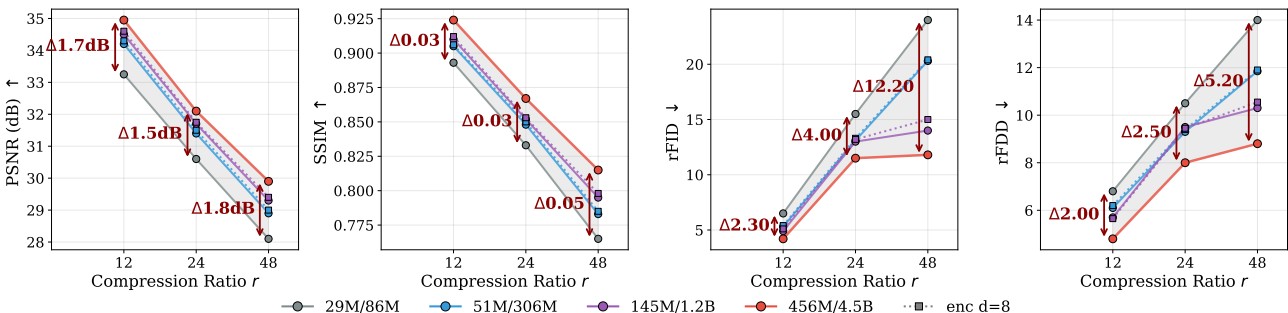

*Figure 2.* **Decoder scaling across compression ratios.** Decoder sizes from B (88M) to T (4.5B) evaluated on ImageNet-1k 256×256 at $r \in \{12, 24, 48\}$. Solid: 4-layer encoder; dotted: 8-layer (minimal difference). Δ: B-to-T gap. Scaling past 350M improves all metrics; the B-to-T rFID gap grows from 2.3 at $r$=12 to 12.2 at $r$=48, motivating the joint scaling study in Section 3.6.

bottleneck imposed by $r$, improving reconstruction without changing the latent space that downstream models learn. To probe this, we train models across compression ratios $r \in \{12, 24, 48\}$ (corresponding to f16×64, f16×32, and f16×16) and scale decoders from 88M to 4.5B parameters.

### 2.1. Architecture

We adopt the ViTok (Hansen-Estruch et al., 2025) architecture with NaFlex-style training (Dehghani et al., 2023; Tschannen et al., 2025) to overcome fixed-resolution limitations. Images are resized preserving aspect ratio, gray-padded to dimensions divisible by patch size $p \in \{16, 32\}$, and patchified into non-overlapping $p \times p$ patches. We use 2D RoPE (Su et al., 2024). We scale decoders across four sizes while keeping encoders shallow (4 layers): B (768 width, 12 depth, 88M params), L (1024 width, 24 depth, 302M), G (1408 width, 40 depth, 1.1B), and T (3072 width, 40 depth, 4.5B). Models are denoted as {enc}-{dec}/{patch}×{ch}, e.g., `Td4-T/16×64` indicates a 4-layer encoder with T-scale decoder, patch size 16, and 64 latent channels. Full architecture details are in Appendix A.1.

### 2.2. Training & Data

We train on ∼2B images from DataComp (Gadre et al., 2023), YFCC-100M (Thomee et al., 2016), Shutterstock, ImageNet-22K (Russakovsky et al., 2015), and LAION (Schuhmann et al., 2022). Following the SigLIP-2 training protocol (Tschannen et al., 2025), training proceeds in two stages: 90% uses 256-token NaFlex, followed by 10% with 1024-token NaFlex for high-resolution generalization (see Appendix B for data processing details). We train with full attention but apply 2D sliding window attention (SWA) only at inference; each query attends to a $(2r+1)^2$ neighborhood in patch space (we use radius $r$=8), reducing complexity from $O(L^2)$ to $O(L \cdot r^2)$.

We use AdamW (Loshchilov & Hutter, 2019) with $\beta = (0.9, 0.95)$, weight decay 0.05, peak learning rate 5 ×

$10^{-4}$ with cosine decay, batch size 8192, and gradient clipping at 1.0. Training uses FSDP in bfloat16 with float8 GEMMs (Micikevicius et al., 2022) and `torch.compile`, achieving ∼90% MFU on 128 H200 GPUs. Full training details are in Appendix B.

### 2.3. Evaluation Metrics

We report reconstruction quality using PSNR and SSIM for pixel fidelity, plus rFID and rFDD for perceptual quality. Both are Fréchet distances (Heusel et al., 2017) computed using Inception-v3 and DINOv3 features, respectively. The "r" prefix distinguishes reconstruction metrics from generation metrics (gFID, gFDD) in Section 3.5. Reconstruction metrics are computed on 50k ImageNet-1k validation images at 256×256 unless otherwise noted. For generation metrics, we generate 100k samples and compare against a 100k reference set from ImageNet-22k training data, constructed to include each class at least once with the remainder sampled randomly.

### 2.4. Losses

We avoid adversarial losses for training stability at scale. Our objective is $\mathcal{L} = \mathcal{L}_{\text{char}} + 0.1\,\mathcal{L}_{\text{SSIM}} + \lambda_p\,\mathcal{L}_{\text{DINO}}$, where $\mathcal{L}_{\text{char}}$ is Charbonnier loss (Charbonnier et al., 1994) for pixel fidelity, $\mathcal{L}_{\text{SSIM}}$ (Wang et al., 2004) preserves structural similarity, and $\mathcal{L}_{\text{DINO}}$ is our DINOv3 perceptual tile loss, building on the DINO self-supervised learning lineage (Caron et al., 2021; Oquab et al., 2023; Siméoni et al., 2025). The weight $\lambda_p$ controls the perception-distortion trade-off: $\lambda_p$=500 balances reconstruction (rFID ≈ 0.7), while $\lambda_p$=1000 prioritizes perceptual metrics (rFID ≈ 0.3) at modest PSNR cost. Loss ablations are in Section 3.4; detailed formulations in Appendix E.

**DINOv3 Perceptual Tile Loss.** Since DINOv3 uses fixed positional embeddings, we sample random 224×224 tiles from corresponding locations in both input and reconstructed images (see Figure 1). Each tile pair is passed through a frozen DINOv3-S encoder (Siméoni et al., 2025);

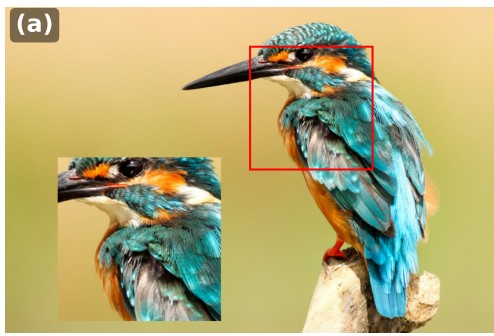
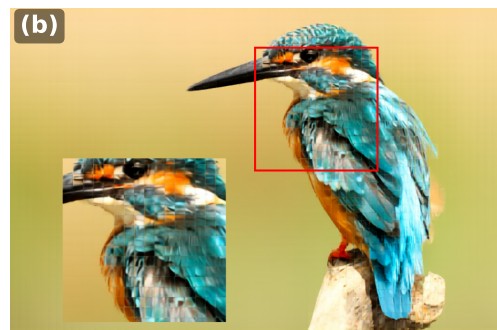
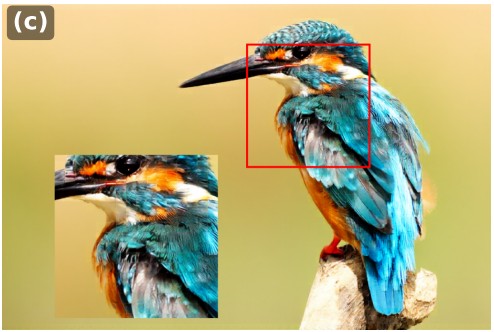
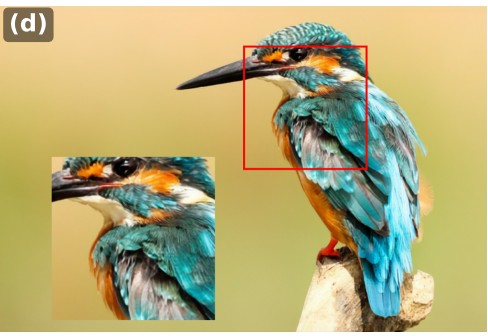

*Figure 3.* **Patch boundary artifacts under different training regimes.** 1024p DIV8K reconstruction with our 5B f16×64 model. (a) Ground truth. (b) Fixed 256×256 crops produce visible grid artifacts. (c) 256-token NaFlex reduces artifacts significantly. (d) 1024-token NaFlex (final 10% of training) removes them entirely. Insets: 2× zoom. The two-stage NaFlex schedule is necessary and sufficient for artifact-free high-resolution reconstruction.

we extract intermediate features from multiple transformer blocks, L2-normalize per token, and compute MSE between feature maps. Gradients flow through DINOv3 to the tokenizer, but DINOv3 weights remain frozen.

# 3. Experiments

We organize our experiments into two parts. First, we ablate the reconstruction autoencoder: encoder depth and width (Section 3.1), decoder scaling across compression ratios $r \in \{12, 24, 48\}$ (Section 3.2), NaFlex training for resolution generalization (Section 3.3), and our DINOv3 loss (Section 3.4). Second, we study downstream generation: latent regularization (Section 3.5), joint AE–flow model scaling (Section 3.6), and the effect of decoder capacity on generation quality (Section 3.7).

## 3.1. Encoder Depth

ViTok demonstrated that decoder capacity matters more than encoder capacity. We push this further by testing extremely shallow encoders. Table 1 shows that reducing from 4 to 1 encoder block has minimal SSIM impact, but a pure linear projection degrades catastrophically (0.462 vs. 0.768 SSIM), demonstrating that non-linear capacity is necessary even for encoding. For f32 compression, Table 1 shows that both encoder and decoder width must match or exceed the

*Table 1.* **Encoder ablations.** *Left (f16):* Reducing encoder depth from 4 to 1 block has minimal impact, but linear projection fails. *Right (f32):* At 32× compression, width must match patch dimensionality (3072). See Appendix A.1 for full model configurations.

| Config | Depth (f16) Enc/Dec | SSIM | Config | Width (f32) Enc/Dec | SSIM |
|--------|---------|------|--------|---------|------|
| Bd4 | 29M/86M | .768 | Bd4-B | 31M/87M | .432 |
| Bd2 | 15M/86M | .766 | Ld4-L | 54M/306M | .688 |
| Bd1 | 8M/86M | .753 | Gd4-G | 149M/1.2B | .793 |
| Linear | 49K/86M | .462 | Td4-T | 463M/4.5B | .872 |

number of pixels per patch token, consistent with width-matching requirements observed in RAE (Zheng et al., 2025) and analogous to DiT findings that transformer width should match latent dimensionality (Peebles & Xie, 2023).

> **Finding 1: Both encoder and decoder width should match or exceed pixels per token, but a 1-block deep encoder is sufficient.**

## 3.2. Decoder Scaling

We compare decoders from sizes B (88M) to T (4.5B) at compression ratios $r \in \{12, 24, 48\}$, with results in Figure 2. Larger decoders consistently improve all metrics, with benefits *increasing* at higher compression: at $r=12$ the B-to-T rFID gap is 2.3, growing to 12.2 at $r=48$. This shows that *decoder capacity is especially critical for highly compressed representations* where the reconstruction task

is more challenging. Scaling past 350M continues to yield improvements, a finding concurrently validated by Giga-Tok (Xiong et al., 2025). Encoder depth (d=4 vs d=8) has minimal impact, consistent with Finding 1 and ViTok-v1.

> **Finding 2: Decoder scaling past 350M benefits all compression rates, with gains increasing at higher $r$.**

### 3.3. NaFlex Training

We ablate three training regimes (see Section 2.1 for setup): (1) **Fixed 256p**: standard 256×256 square crops; (2) **256-token NaFlex**: variable aspect ratios fitting 256 token budget; (3) **1024-token NaFlex**: 1024 token budget for the final 10% of training (Tschannen et al., 2025).

We train with full attention but apply SWA only at inference with negligible quality degradation, implying that attention heads gather useful information primarily from local tokens even when trained with global access. Figure 3 illustrates the progression: fixed 256p training produces visible grid artifacts at non-square resolutions, 256-token NaFlex significantly reduces these artifacts by exposing the model to varied aspect ratios, and the 1024-token NaFlex stage (final 10% of training) eliminates them entirely. This two-stage schedule is critical for clean high-resolution reconstruction while keeping most of training at the cheaper 256-token budget.

> **Finding 3: NaFlex enables resolution generalization; SWA can be added at inference to tokenizers trained with full attention without performance degradation.**

### 3.4. Loss Ablation

*Table 2.* **Loss ablation on ImageNet-1k 256×256.** Each "+" row adds one loss component to the pixel-only baseline. Model: 5B Td4-T/16×64, $r$=12. Evaluated on 50k val images. rFID/rFDD use Inception-v3/DINOv3 features. Green: improvement; red: degradation vs. pixel-only.

| Training Style | rFID↓ | rFDD↓ | SSIM↑ |
|---|---|---|---|
| Pixel-only | 5.13 | 10.96 | **.929** |
| +SSIM | 4.87↓5% | 10.42↓5% | .931↑0.2% |
| +LPIPS | 0.72↓86% | 2.93↓73% | .923↓0.6% |
| +DINO ($\lambda$=250) | 0.51↓90% | 1.45↓87% | .919↓1.1% |
| +DINO ($\lambda$=1000) | **0.30**↓94% | **1.12**↓90% | .914↓1.6% |
| +LPIPS+DINO | 0.38↓93% | 1.35↓88% | .918↓1.2% |

We ablate our loss components: Charbonnier for pixel fidelity, SSIM for structural similarity, LPIPS (Zhang et al., 2018) for VGG-based perceptual quality, and our DINOv3 perceptual tile loss (Caron et al., 2021; Oquab et al., 2023; Siméoni et al., 2025) for self-supervised patch-level features. Table 2 shows results on ImageNet 256×256 with all experiments run with our 5B decoder model.

Pixel-level losses achieve highest PSNR (34.81 dB) but worst perceptual metrics (rFID 5.13). LPIPS reduces rFID

by 86% to 0.72. Our DINOv3 loss achieves strictly better perceptual improvements: $\lambda_{dino}$=1000 yields rFID 0.30, a 58% improvement over LPIPS, with comparable SSIM cost. We attribute this to DINOv3's self-supervised patch-level training, which preserves fine-grained spatial information while capturing semantic structure. In contrast, LPIPS's VGG features are trained for classification and are invariant to reconstruction-relevant details. Combining LPIPS with DINO provides no additional benefit (rFID 0.38 vs. 0.30 DINO-only), suggesting that DINO subsumes the perceptual signal provided by LPIPS.

> **Finding 4: Our DINOv3 loss is a stronger perceptual loss than LPIPS, notably improving rFID and rFDD with comparable SSIM.**

### 3.5. Latent Regularization

*Table 3.* **Latent regularization comparison.** KL ($\beta$=0.01), tanh+noise ($\sigma$=0.01), and LayerNorm evaluated with a 450M flow transformer on ImageNet-1k 256p (100 epochs, 5B f16×64 AE). All achieve similar gFID (4.7–5.0); we adopt LayerNorm for simplicity.

| Method | rFID↓ | gFID↓ | gFDD↓ | IS↑ | Std |
|---|---|---|---|---|---|
| KL ($\beta$=0.01) | 0.43 | **4.71** | 5.32 | 181 | 0.39 |
| Tanh+Noise | 0.42 | 5.02 | 5.42 | 178 | 1.04 |
| LayerNorm | 0.43 | 4.73 | **5.24** | **185** | 1.00 |

Having established our reconstruction recipe, we now turn to downstream generation. Recent work suggests that scaling to higher latent channels requires sophisticated latent space structuring, such as high-frequency emphasis (Skorokhodov et al., 2025) or DINO alignment (Yao & Wang, 2025; Zheng et al., 2025). We investigate whether simpler approaches suffice, starting with the choice of latent regularization. We use the flow model training setup described in Appendix C and A.2.

We compare three approaches: KL divergence with $\beta$=0.01 (proper VAE), tanh bounding with scaled Gaussian noise ($\sigma$=0.01), and LayerNorm (fully deterministic). As shown in Table 3, all methods achieve similar gFID (4.7–5.0) and rFID, indicating that for high-channel latents the choice of stochasticity has minimal impact. This is consistent with DiTo (Chen et al., 2025c), which also adopts LayerNorm. We choose LayerNorm for simplicity: it requires no hyperparameter tuning and is fully deterministic.

> **Finding 5: High-channel latent generation is largely insensitive to the choice of latent regularization.**

### 3.6. Joint AE–Flow Model Scaling

To control for compute, we plot generation metrics against cumulative flow model training FLOPs in Figure 4. We train DiT-style flow transformers (Peebles & Xie, 2023) at

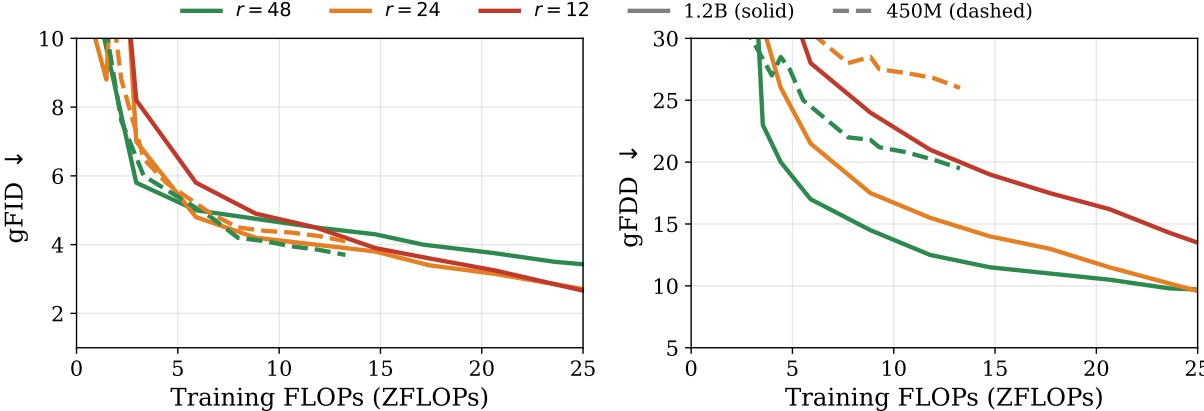

Figure 4. **Joint AE–flow model scaling on ImageNet-22k.** DiT-style flow transformers (450M dashed, 1.2B solid) trained for 300 epochs at $r \in \{12, 24, 48\}$. X-axis: cumulative FLOPs. **(a)** gFID: the 450M flow model performs best at $r{=}48$ (highest compression); the 1.2B model makes lower-$r$ latents competitive, exploiting the richer representations that Figure 2 shows scale most from decoder capacity. **(b)** gFDD: larger models consistently outperform across all $r$.

two scales (450M and 1.2B) for 300 epochs on ImageNet-22k, using latents from our 5B AE at compression ratios $r \in \{12, 24, 48\}$.

For gFID (Inception-v3 features), the 450M flow transformer at $r{=}48$ (highest compression) outperforms the lower-$r$ variants at matched FLOPs, consistent with a more compact latent space being easier for smaller generators to learn. However, this ranking flips with the 1.2B flow transformer: $r{=}12$ and $r{=}24$ become competitive or superior, indicating that larger flow models can better exploit the richer representations available at lower compression. For gFDD (DINOv3 features), larger flow models consistently outperform smaller ones across all compression ratios at matched FLOPs. Interestingly, gFID and gFDD rankings diverge: $r{=}12$ achieves competitive gFID with the 1.2B flow transformer but lags in gFDD, suggesting that Inception-v3 and DINOv3 capture different aspects of generation quality and that low-compression latents may still pose challenges for semantic fidelity as measured by DINOv3.

### 3.7. Decoder Capacity

To isolate the effect of decoder capacity on downstream generation, we train 1.2B flow transformers on ImageNet-22k using latents from two AE sizes: our 5B (Td4-T) and 350M (Ld4-L) decoders. Figure 5 plots reconstruction metrics (rFID, rFDD on 50k ImageNet-1k val) against generation metrics (gFID, gFDD on 50k samples) for both AE sizes across compression ratios $r \in \{12, 24, 48\}$ at 256p (f16). We also include a 512p ablation using the f32 configuration.

Across all compression ratios and resolutions, the 5B AE (red) achieves 1–2 gFID points better generation quality than the 350M AE (blue). This holds even when the 5B AE shows similar or worse rFID, indicating that decoder capacity benefits generation through mechanisms beyond

reconstruction fidelity alone. Combined with the flow model scaling results in Section 3.6, both decoder capacity and flow model capacity independently contribute to unlocking the benefits of lower compression ratios.

> **Finding 6: Larger generators can effectively utilize autoencoders with low compression ratio $r$.**

## 4. Comparison

We compare ViTok-v2 against prior visual tokenizers, both CNN-based (SD-VAE (Rombach et al., 2022), SDXL-VAE (Podell et al., 2023), FLUX (Black Forest Labs, 2024), DC-AE (Chen et al., 2024), Qwen, Hunyuan (Kong et al., 2024)) and ViT-based (VA-VAE, RAE (Zheng et al., 2025), AToken (Lu et al., 2025)), on reconstruction benchmarks spanning 256p to 8192p. Evaluation metrics (rFID, rFDD, PSNR, SSIM) and methodology follow Sections 2.3 and 2.1 respectively.

### 4.1. 256p and 512p Reconstruction

Table 4 summarizes reconstruction quality across resolutions and compression ratios. ViTok-v2 5B consistently achieves the best PSNR and SSIM at each compression ratio, demonstrating superior pixel-level fidelity. For perceptual metrics (rFID/rFDD), ViTok-v2 is competitive, generally matching or beating baselines at 512p, while FLUX.2 leads at 256p due to its adversarial training. This rFID gap can be closed: increasing our DINO loss weight ($\lambda{=}1000$) yields rFID 0.30 on ImageNet 256p, approaching FLUX.2's 0.15, with only modest PSNR reduction (see ‡ rows).

At $r{=}12$, ViTok-v2 5B achieves +3.1 dB PSNR over FLUX.2 on ImageNet 256p; this gap widens to 2–3 dB at $r{=}24$ and $r{=}48$. The high-DINO variant ([‡]) demonstrates the perception-distortion trade-off: rFID drops from

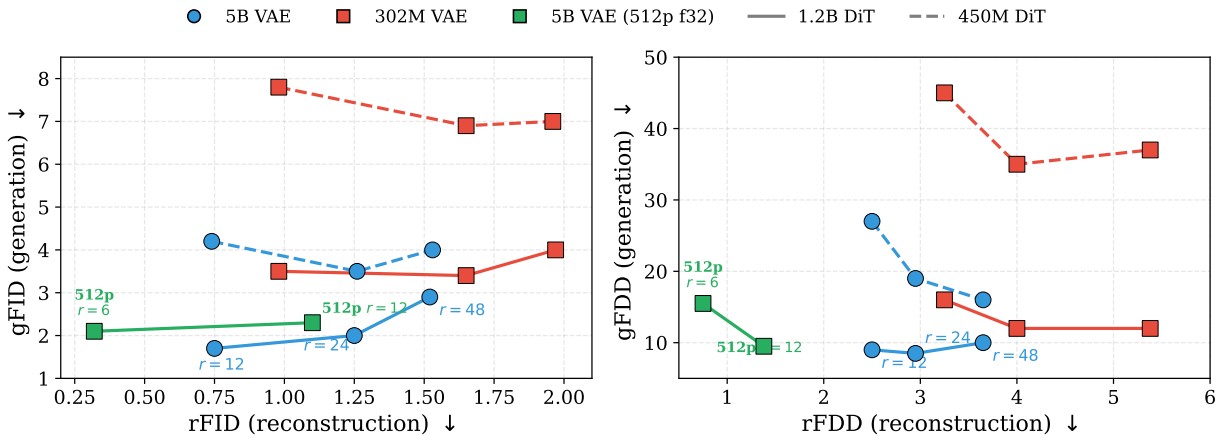

Figure 5. **Reconstruction vs. generation quality.** Reconstruction metrics (rFID, rFDD) vs. generation metrics (gFID, gFDD) for 5B AE (red) and 350M AE (blue) at $r \in \{12, 24, 48\}$, 256p. Points are annotated with $r$ values. **(a)** rFID vs. gFID. **(b)** rFDD vs. gFDD. The 5B AE achieves 1–2 gFID improvement over 350M even at comparable reconstruction quality, confirming decoder capacity benefits generation beyond reconstruction fidelity.

Table 4. **Reconstruction quality on ImageNet 256p, COCO 256p, and COCO 512p.** ViTok-v2 5B vs. CNN- and ViT-based methods at $r \in \{12, 24, 48\}$. † from original papers; ‡ high DINO loss weight ($\lambda{=}1000$) trading PSNR for better rFID.

| Model | Comp. | $r$ | ImageNet 256p | | | | COCO 256p | | | | COCO 512p | | | |
|---|---|---|---|---|---|---|---|---|---|---|---|---|---|---|
| | | | FID↓ | FDD↓ | PSNR↑ | SSIM↑ | FID↓ | FDD↓ | PSNR↑ | SSIM↑ | FID↓ | FDD↓ | PSNR↑ | SSIM↑ |
| FLUX.2 | f8×16 | 12 | .15 | – | 31.1 | .887 | **1.12** | **1.46** | 31.5 | .900 | .45 | .45 | 33.2 | .914 |
| FLUX.1 | f8×16 | 12 | .24 | 1.26 | 31.1 | .887 | – | – | – | – | – | – | – | – |
| Hunyuan | f8×16 | 12 | .67 | – | 33.3 | .916 | – | – | – | – | – | – | – | – |
| VA-VAE | f16×64 | 12 | **.14** | – | 30.7 | .870 | – | – | – | – | – | – | – | – |
| Qwen | f8×16 | 12 | 1.32 | 7.36 | 30.3 | .860 | 1.71 | 3.79 | 29.1 | .849 | .79 | 2.24 | 30.8 | .863 |
| ViTok-v2 5B | f16×64 | 12 | .74 | 2.49 | **34.2** | **.924** | 2.98 | 4.28 | **34.1** | **.930** | .41 | 1.02 | **35.9** | **.938** |
| ViTok-v2 5B‡ | f16×64 | 12 | .30 | **1.12** | 33.6 | .914 | 1.85 | 2.15 | 33.5 | .921 | **.38** | **.72** | 32.1 | .925 |
| SDXL-VAE† | f8×8 | 24 | .68 | – | 26.0 | .834 | **4.16** | 9.59 | 25.8 | .740 | 2.20 | 4.77 | 27.5 | .762 |
| AToken† | f16×32 | 24 | **.26** | – | 28.8 | .814 | – | – | – | – | – | – | – | – |
| VA-VAE | f16×32 | 24 | .28 | – | 28.0 | .790 | – | – | – | – | – | – | – | – |
| ViTok-v2 5B | f16×32 | 24 | 1.26 | **2.94** | 31.2 | .867 | 4.72 | **5.37** | 31.1 | .878 | 1.02 | 1.44 | 32.9 | .888 |
| SD-VAE | f8×4 | 48 | .73 | 6.14 | 25.7 | .702 | **4.38** | 12.1 | 25.4 | .715 | **2.28** | 4.78 | 27.1 | .742 |
| VA-VAE | f16×16 | 48 | **.55** | – | 25.3 | .690 | – | – | – | – | – | – | – | – |
| RAE | f16×16 | 48 | .61 | – | 18.8 | .496 | – | – | – | – | – | – | – | – |
| ViTok-v2 5B | f16×16 | 48 | 1.52 | **3.66** | 28.5 | .793 | 6.66 | **7.15** | 28.3 | .807 | 2.27 | 2.24 | 30.1 | .820 |

0.74 to 0.30 (approaching FLUX.2's 0.15) at the cost of 0.6 dB PSNR. At 512p, ViTok-v2 achieves best or near-best results across all metrics; sliding window attention enables fast, high-quality reconstruction where CNN-based methods exhibit significant latency.

### 4.2. High-Resolution Reconstruction

Table 5 shows high-resolution results on DIV8K. ViTok-v2 substantially outperforms all baselines: at 2048p, we achieve rFID 0.06 (87% better than FLUX.2) with +2.3 dB PSNR. The efficiency gap is even more striking at higher resolutions. CNN-based methods either time out (>60s/image) or run out of memory at 4K+, while the transformer architecture with sliding window attention scales gracefully: ViTok-v2 5B processes 4K in 1.2s (**50×+ faster**) and f32

in 0.4s. At 8192p, only ViTok-v2 succeeds; all baselines OOM.

The f16 and f32 configurations offer a practical quality–speed trade-off: f32×128 processes 4K images in 0.4s (3× faster than f16×64) at $r{=}24$, trading ∼2 dB PSNR for substantially lower latency. This makes the f32 variants attractive for latency-sensitive pipelines, while f16 remains preferable when reconstruction fidelity is paramount. Critically, the ability to process 8K images at all, where every CNN baseline we tested runs out of memory, opens new applications for high-resolution generation and editing workflows that were previously intractable with learned tokenizers.

*Table 5.* **High-resolution reconstruction on DIV8K (1024p–8192p).** CNN methods timeout or OOM at 4K+. ViTok-v2 with SWA scales linearly: 4K in 0.4–1.2s, 8K in 1.3–5.7s.

| Model | Comp. | $r$ | DIV8K 1024p | | | | | DIV8K 2048p | | | | | Latency | |
| | | | FID↓ | FDD↓ | PSNR↑ | SSIM↑ | ms | FID↓ | FDD↓ | PSNR↑ | SSIM↑ | ms | 4K | 8K |
|---|---|---|---|---|---|---|---|---|---|---|---|---|---|---|
| FLUX.2 | f8×16 | 12 | .90 | **.44** | 31.4 | .908 | 108 | .48 | .16 | 32.6 | .918 | 284 | >60s | OOM |
| Qwen | f8×16 | 12 | 1.50 | 1.28 | 28.8 | .845 | 237 | .69 | .31 | 30.0 | .864 | 790 | OOM | OOM |
| ViTok-v2 5B | f16×64 | 12 | **.35** | .89 | **34.0** | **.932** | 207 | **.06** | **.11** | **34.9** | **.938** | 294 | 1.2s | 5.4s |
| ViTok-v2 5B | f16×32 | 24 | **1.21** | **2.16** | **31.0** | **.874** | 71 | **.19** | **.47** | **32.0** | **.886** | 292 | 1.2s | 5.2s |
| ViTok-v2 5B | f32×128 | 24 | 1.68 | 3.13 | 29.1 | .834 | **15** | .73 | .70 | 30.0 | .853 | **70** | **.4s** | **1.3s** |
| SD-VAE | f8×4 | 48 | 5.59 | 4.38 | 25.6 | .707 | 110 | 1.92 | 1.91 | 26.6 | .738 | 290 | >60s | OOM |
| ViTok-v2 5B | f16×16 | 48 | **3.05** | **3.17** | **28.5** | **.802** | 71 | **.69** | **1.11** | **29.4** | **.818** | 290 | 1.2s | 5.7s |
| ViTok-v2 5B | f32×64 | 48 | 4.69 | 6.14 | 27.0 | .754 | **15** | .67 | 1.87 | 28.1 | .782 | **71** | **.5s** | **1.3s** |
| DC-AE-f32 | f32×32 | 96 | 7.08 | 4.88 | 23.7 | .636 | 335 | – | – | – | – | OOM | OOM | OOM |
| DC-AE-f64 | f64×128 | 96 | 6.52 | 4.33 | 24.1 | .648 | 422 | – | – | – | – | OOM | OOM | OOM |

## 5. Related Work

**Visual tokenization.** Deep autoencoders compress images into continuous latent representations for efficient generative modeling (Kingma & Welling, 2013; Oord et al., 2017). SD-VAE (Rombach et al., 2022) established the standard architecture for latent diffusion: a convolutional encoder-decoder with 8× spatial compression and 4 channels, producing 1024 latent tokens per 256px image. SDXL-VAE improved reconstruction quality via larger training batches and longer schedules (Podell et al., 2023), while Flux (Black Forest Labs, 2024) extends to 16 channels with adversarial training for sharper reconstructions. These CNN-based AEs benefit from translation invariance, enabling robust cross-resolution generalization even when trained only at 256px.

**ViT-based autoencoders.** Vision Transformer autoencoders offer an alternative to CNNs with more aggressive compression and cleaner scaling properties. ViTok (Hansen-Estruch et al., 2025) demonstrated that decoder capacity determines reconstruction quality more than encoder capacity, matching CNN-AE fidelity while using only 256 tokens (vs. 1024), enabling 4× faster diffusion training. GigaTok (Xiong et al., 2025) scaled discrete tokenizers to 3B parameters using DINOv2 regularization. TiTok (Yu et al., 2024) achieves extreme compression to just 32 tokens using learned queries. AToken (Lu et al., 2025) unifies image, video, and 3D tokenization with 4D RoPE and adversarial-free training. FlowMo (Sargent et al., 2025) replaces adversarial losses with a diffusion decoder, achieving state-of-the-art reconstruction without convolutions or 2D latent codes. MAE-Tok (Chen et al., 2025b) shows that masked autoencoder pretraining produces better latent structure than variational constraints. However, vanilla ViTs struggle at resolutions beyond their training distribution: models trained at 256px exhibit severe grid artifacts at 512px due to positional embedding extrapolation failures.

**High-compression & flexible-resolution.** DC-AE (Chen et al., 2024) achieves extreme 32–128× compression ratios for efficient 4K generation (Xie et al., 2024); DC-AE 1.5 (Chen et al., 2025a) extends this with channel masking for improved downstream generation. Scale-RAE (Tong et al., 2026) scales regularized autoencoders to large-scale text-to-image generation, showing that VAE-based models overfit while RAE models remain stable. NaViT (Dehghani et al., 2023) introduced native aspect ratio training for ViTs, later extended by SigLIP-2 (Tschannen et al., 2025) and FiT (Lu et al., 2024) for flexible-resolution diffusion. We adopt NaFlex-style training combined with 2D RoPE to enable resolution generalization in ViT-AEs.

**Perceptual losses & trade-offs.** LPIPS (Zhang et al., 2018) uses pretrained VGG features to measure perceptual similarity, while adversarial training with GANs (Esser et al., 2021) produces sharper outputs at the cost of training instability. DINOv3 (Siméoni et al., 2025) provides modern self-supervised features with strong spatial correspondence; SVG (Shi et al., 2025) goes further by replacing the VAE entirely with frozen DINO features for 62× faster training. We demonstrate DINOv3 can replace both LPIPS and GAN objectives while enabling stable billion-parameter training. The perception-distortion trade-off (Blau & Michaeli, 2018) is well-documented: optimizing perceptual metrics (FID) often degrades pixel fidelity (PSNR). Additionally, a reconstruction-generation trade-off exists where larger bottlenecks improve reconstruction but can hurt downstream generation quality (Hansen-Estruch et al., 2025); we systematically study both trade-offs in Sections 3.4 and 3.5.

## 6. Conclusion

We presented ViTok-v2, a family of ViT-based visual tokenizers scaled to 5B parameters. Key findings: (1) Encoder depth matters less than width; (2) NaFlex training in two stages, with the first 90% of training using a 256-token budget then the last 10% using a 1024-token budget, enables high-resolution reconstruction that generalizes to

natural image resolutions and aspect ratios; (3) Our novel DINOv3 loss matches LPIPS's performance, while enabling stable GAN-free training; (4) Decoder scaling benefits increase with higher compression, but the encoder size has minimal impact on performance; (5) Larger generation models better leverage high-channel latents and larger autoencoders enable better generation quality even if reconstruction performance is worse. ViTok-v2 achieves state-of-the-art PSNR/SSIM on COCO and DIV8K with +3 dB over FLUX.2. Additionally, our ViT-based approach leveraging sliding window attention enables the processing of 8k images, where all CNN-based tokenizers are unusable due to running out of memory.

## Impact Statement

This paper presents work to advance visual tokenization for image generation. Improved tokenizers may enable more accessible image synthesis, with associated risks of misuse for generating misleading visual content. We acknowledge that higher-fidelity reconstruction and generation could lower the barrier for producing deceptive imagery. To mitigate this, we release models and code to facilitate research into detection and attribution methods, and we encourage the community to develop robust provenance tracking alongside generative capabilities.

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

# A. Architecture Details

## A.1. ViTok-v2 Autoencoder

**Model Variants.**   We scale decoders from 88M to 4.5B parameters by increasing width (Table 6). Encoders use 4 layers with matched width across all variants.

*Table 6.* **Model variants.**

| Scale | Width | Depth | Heads | Params |
|-------|-------|-------|-------|--------|
| B | 768 | 12 | 12 | 88M |
| L | 1024 | 24 | 16 | 302M |
| G | 1408 | 40 | 16 | 1.1B |
| T | 3072 | 40 | 24 | 4.5B |

**Encoder Design.**   A shallow stack (depth 4, width matching decoder) maps input patches to latents via a linear head. We use a non-variational design with tanh-bounded output and small i.i.d. Gaussian noise during training. The encoder comprises only ∼10% of total parameters, validating our asymmetric architecture where decoder capacity dominates.

**Decoder Design.**   Latents are projected to hidden dimension and processed by a deep transformer stack. A final linear layer predicts per-token RGB patches, which are unpatchified to reconstruct the image.

**Attention and Positional Encoding.**   We use pre-norm transformer blocks with multi-head self-attention (per-head Q/K RMSNorm) and 2D rotary positional embeddings (RoPE). The MLP uses SwiGLU with expansion ≈2.67. LayerScale stabilizes residual branches. For high-resolution inference, sliding window attention (SWA) with 2D block radius 8 enables linear memory scaling.

## A.2. Transformer-Based Flow Model

Our flow model operates directly on ViTok-v2 latents $z \in \mathbb{R}^{B \times L \times c}$, following the DiT architecture (Peebles & Xie, 2023) with modern improvements.

**Architecture.**   Latents are projected to hidden dimension $D$ with learned absolute position embeddings. Each block applies: (1) self-attention on affinely modulated pre-norm, (2) cross-attention to text/class conditioning, and (3) SwiGLU MLP, with LayerScale on all residuals. We use AdaLN (Peebles & Xie, 2023) for timestep modulation and 2D RoPE (Su et al., 2024) for positional encoding.

**Output Head.**   The final head maps hidden states back to latent dimension to predict flow matching targets in the ViTok-v2 latent space.

# B. Pretraining Setup

**Training Data.**   We train on ∼2B images from DataComp, YFCC-100M, Shutterstock, ImageNet-22K, and LAION, streamed via WebDataset shards.

**NaFlex Variable-Resolution Training.**   Unlike fixed-resolution training that resizes all images to a canonical size (e.g., 256×256), we use NaFlex (Dehghani et al., 2023) to process images at their native aspect ratios within a *token budget*. Each image is randomly cropped and resized to fit the sampled budget while preserving aspect ratio, exposing the model to diverse resolutions during training. Training proceeds in two stages: 90% of training uses a 256-token budget (∼256p at patch size 16), followed by 10% with a 1024-token budget (∼512p) to enable high-resolution generalization. Variable-length sequences are packed with boolean attention masks for efficient batching.

**Hyperparameters.**   AdamW ($\beta = (0.9, 0.95)$, weight decay 0.05, lr $5 \times 10^{-4}$) with cosine schedule and gradient clip 1.0 for ∼300k steps. Batch size 8192 with gradient accumulation. Training uses FSDP in bfloat16. Through bfloat16, float8 GEMMs (Micikevicius et al., 2022), and `torch.compile`, we achieve 90% MFU on 128 H200s.

**Loss Weights.** We use $\lambda_{\text{char}} = 1.0$, $\lambda_{\text{SSIM}} = 0.1$, and $\lambda_{\text{dino}} = 500$ (default) or $\lambda_{\text{dino}} = 1000$ (high-perceptual variant). We do not use adversarial losses, which simplifies training and avoids mode collapse while still achieving strong perceptual quality through the DINOv3 loss.

## C. Flow Model Training

To evaluate generation quality (Section 3.5), we train transformer-based flow models on ViTok-v2 latents using flow matching.

**Model Configurations.** We train two scales: DiT-450M (depth 24, width 1152, 16 heads) and DiT-1.2B (depth 32, width 1536, 24 heads). Both use SwiGLU MLPs, 2D RoPE, and AdaLN for conditioning.

**Training Setup.** Models are trained on ImageNet-22k ($10\times$ larger than ImageNet-1k for rigorous generalization testing) for 300 epochs with batch size 4096, EMA decay 0.999, and cosine LR schedule with peak $1 \times 10^{-3}$. We use Euler sampling with 50 steps and classifier-free guidance scale 4.0 at inference.

## D. Evaluation Protocol

**Metrics.** We evaluate reconstruction quality using four complementary metrics:

- **rFID** (reconstruction FID): Fréchet Inception Distance between ground truth and reconstructed images using Inception-v3 features. Lower is better.
- **rFDD** (reconstruction FDD): Fréchet DINOv3 Distance using DINOv3 features, capturing semantic similarity. Lower is better.
- **PSNR**: Peak Signal-to-Noise Ratio measuring pixel-level fidelity. Higher is better.
- **SSIM**: Structural Similarity Index measuring perceptual similarity via luminance, contrast, and structure. Higher is better.

**Datasets and Sampling.** For COCO evaluation, we use 5,000 randomly sampled validation images with ADM-style center cropping (Dhariwal & Nichol, 2021) to the target resolution. For DIV8K, we evaluate on the full validation set at native resolution. ImageNet evaluations use the standard 50K validation split.

**Inference Configuration.** All evaluations use float8 inference with `torch.compile` on a single NVIDIA H100 GPU. We use deterministic sampling (no stochastic augmentation at inference) for reproducibility.

## E. Loss Functions

We train without adversarial losses to avoid instability. Our objective combines:

**Charbonnier Loss.** $\mathcal{L}_{\text{char}} = \sqrt{(x - \hat{x})^2 + \epsilon^2}$ with $\epsilon = 10^{-3}$. A smooth L1 approximation providing stable gradients near zero while being robust to outliers.

**SSIM Loss.** Structural similarity (Wang et al., 2004) computed over local windows, capturing luminance, contrast, and structure. We use $\mathcal{L}_{\text{SSIM}} = 1 - \text{SSIM}(x, \hat{x})$.

**DINOv3 Perceptual Tile Loss.** Following Sauer et al. (2024), we use frozen DINOv3-S (Siméoni et al., 2025) features rather than VGG-based LPIPS. We sample random 224×224 tiles from corresponding locations in input and reconstruction, extract intermediate features from multiple transformer blocks, L2-normalize token-wise, and compute MSE. Since DINOv3 uses fixed positional embeddings, this tile-based approach enables perceptual supervision at any resolution.

## F. Trade-off Visualizations

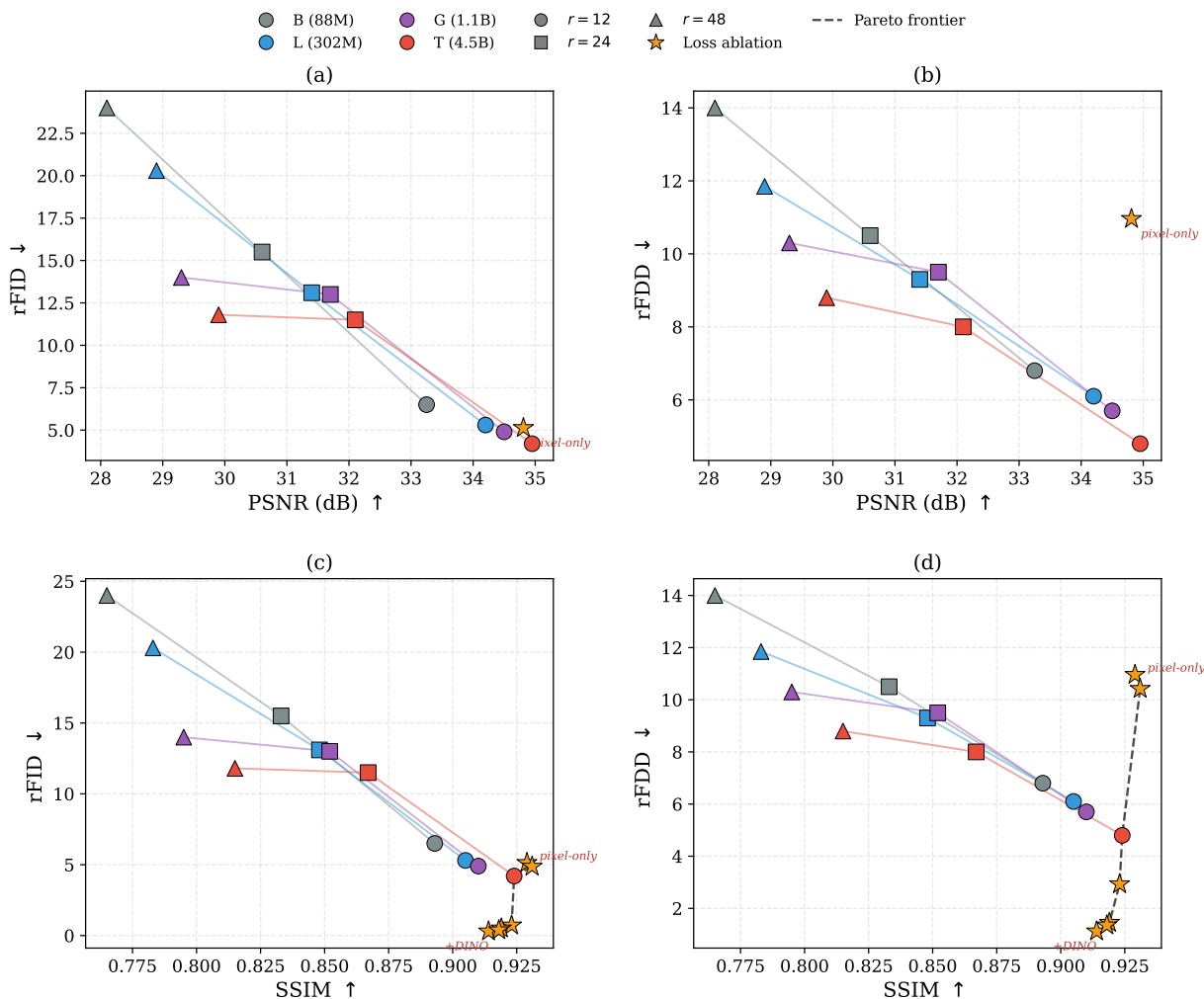

*Figure 6.* **Perception-distortion trade-off with Pareto frontier.** 2×2 grid: columns show rFID and rFDD; rows show PSNR and SSIM on the $x$-axis. Colored circles/squares/triangles are decoder scaling variants (B/L/G/T $\times$ $r \in \{12, 24, 48\}$); orange stars are loss ablation configurations (Table 2, T decoder, $r$=12). The dashed line is the Pareto frontier. **(a, b)** Decoder scaling at fixed loss improves both pixel fidelity and perceptual quality simultaneously, moving points along the frontier. **(c, d)** Loss ablation at fixed decoder creates a nearly *vertical* trade-off: rFID spans 0.3–5.1 within a narrow SSIM band (.914–.931), showing that perceptual losses trade minimal distortion for large perceptual gains. Combining both levers (large decoder + DINOv3 loss) reaches a region inaccessible to either alone.

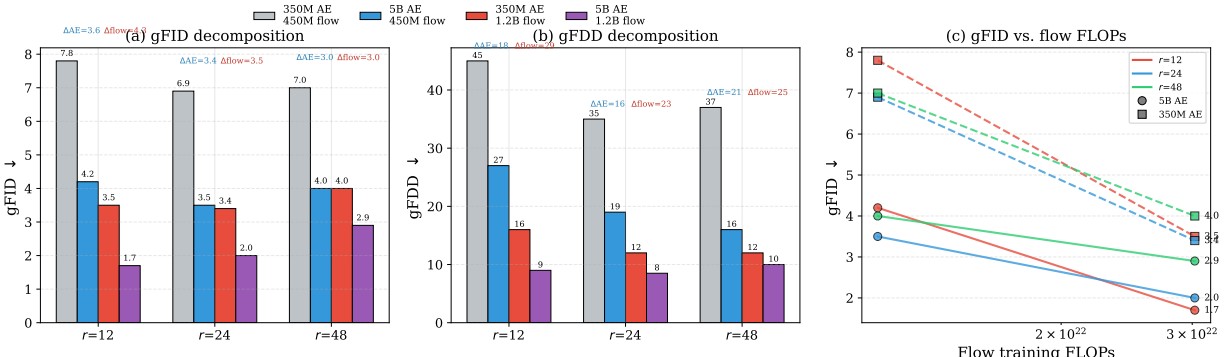

*Figure 7.* **Scaling decomposition of generation quality. (a, b)** Grouped bars decompose gFID and gFDD improvements into AE scaling (350M→5B) and flow scaling (450M→1.2B) contributions. At $r$=12, flow scaling dominates ($\Delta$gFID = 4.3 vs. 3.6 for AE); at higher $r$ the contributions converge. **(c)** gFID vs. flow training FLOPs. The 5B AE (solid) provides a consistent downward shift relative to the 350M AE (dashed) at all FLOPs budgets—an effectively free quality gain since the AE is trained once and amortized across flow model scales and sampling budgets.

