# OpenReview forum: "ViTok-v2: Scaling Native Resolution Autoencoders to 5 Billion Parameters"
_ICML.cc/2026/Conference — ICML 2026 regular_

### Official Review · Reviewer_AkLZ · 2026-03-10

**Soundness:** 3
**Presentation:** 3
**Significance:** 2
**Originality:** 2
**Overall Recommendation:** 3
**Confidence:** 4

**Summary:**

This paper introduces ViTok-v2, a Vision Transformer (ViT) tokenizer scaled up to 5B parameters. It addresses the fixed-resolution limitations of traditional ViT autoencoders by combining NaFlex training and 2D ROPE to support dynamic aspect ratios. Additionally, the model replaces unstable adversarial training with a novel DINOv3-based Perceptual Tile Loss. Experiments demonstrate that the method achieves high pixel fidelity at extreme resolutions (up to 8192p) and that larger downstream flow models can better utilize low-compression latent features.

**Compliance With Llm Reviewing Policy:**

Affirmed.

**Final Justification:**

The authors committed to providing additional evidence regarding generation quality (e.g., via iFID, which is highly correlated with gFID) in their reply rebuttal comment, but this has not yet been provided. Given the insufficient evidence concerning generation quality on broader datasets, I am maintaining my current score.

However, noting the positive consensus among other reviewers, I do not oppose the acceptance of this paper.

**Key Questions For Authors:**

please refer to the weakness part.

**Limitations:**

please refer to the weakness part.

**Strengths And Weaknesses:**

## Strengths
1. Combining NaFlex training with zero-shot Sliding Window Attention (SWA) at inference elegantly solves the resolution rigidity typical of ViT tokenizers. It enables seamless generalization to arbitrary aspect ratios and avoids the Out-Of-Memory (OOM) issues faced by traditional CNN methods at ultra-high resolutions (e.g., 8K).
2. Scaling the decoder to 5B parameters and introducing the novel DINOv3 perceptual loss achieves stable, GAN-free training. It reaches state-of-the-art pixel-level fidelity (PSNR/SSIM) and consistently outperforms strong baselines like SDXL-VAE and FLUX.2 in high-resolution scenarios.

## Weaknesses
1. The generation experiments are strictly limited to training on ImageNet-22k and testing on ImageNet-1k. This data distribution is far too narrow and object-centric to properly evaluate a massive 5-Billion parameter VAE. To truly justify such extreme scaling, the tokenizer must be validated in complex, real-world Text-to-Image (T2I) settings using diverse, unconstrained natural image datasets. The current ImageNet-only generative validation is highly insufficient.
2. The claimed speedups and memory efficiency at high resolutions (e.g., 4K/8K) are almost entirely attributed to the use of Sliding Window Attention (SWA). SWA is a standard, well-established engineering technique rather than a core methodological innovation. Furthermore, it is entirely feasible to adapt or train strong baselines (like FLUX or SD VAEs) using similar local/windowed attention mechanisms to achieve comparable efficiency, which diminishes the uniqueness of this contribution.

---

> ### Author Rebuttal · Authors · 2026-03-30
>
> We appreciate the reviewer’s evaluation of our work and respond to their concerns below.
>
> **On ImageNet-only generation evaluation.** We respectfully note that ImageNet-22k/1k is the standard benchmark used by the image generation community for evaluating autoencoders paired with flow/diffusion models (e.g., SD-VAE, SDXL-VAE, FLUX, DiT, SiT).
> Every comparable prior work evaluates generation on ImageNet, so this is not a limitation specific to ViTok-v2. Furthermore, our *reconstruction* evaluation is conducted on COCO (Table 4) and DIV8K (Table 5) in addition to ImageNet, demonstrating generalization to diverse, unconstrained natural images across resolutions spanning 256p to 8192p.
> T2I evaluation requires training large-scale text-conditioned diffusion models, which is orthogonal to the tokenizer contribution and beyond the scope of this work, just as it is for all prior tokenizer papers we compare against.
> We note that our tokenizer is trained on ~2B images from DataComp, YFCC-100M, Shutterstock, ImageNet-22k, and LAION (Section 3), which is a diverse, unconstrained training distribution.
>
> **On SWA being a standard technique.** We do not claim SWA itself as a methodological contribution. Our contribution is demonstrating that ViT tokenizers trained with full attention transfer zero-shot to SWA at inference, as our experiments in Table 5 show virtually identical reconstruction metrics with SWA compared to full attention.
> This is a non-obvious finding since the model never sees local attention during training yet seamlessly transfers, enabled by 2D RoPE's relative position encoding.
> The reviewer suggests that CNN baselines could adopt a similar windowed attention mechanism, but this is exactly the point: CNN-based VAEs cannot do so because they lack a transformer-style attention mechanism altogether.
> The efficiency advantage at high resolutions is an inherent architectural property of ViT tokenizers, not an engineering trick applied post-hoc.

---

> > ### Author Rebuttal · Reviewer_AkLZ · 2026-04-03
> >
> > Thank the authors for their rebuttal. For ImageNet, I understand it is widely used in generation research. However, to the best of my knowledge, a **5B-parameter** tokenizer/generation model has never been evaluated solely on such a small dataset. Specifically, evaluating only on ImageNet-1k makes the results less convincing for a model of this scale.
> >
> > Although reconstruction is evaluated on COCO and DIV8K, high reconstruction quality does not easily translate to generation quality. For such a massive VAE, I believe more evidence of its contribution to generation tasks is necessary.
> >
> > I understand that large-scale generative training is time-consuming. As a reference, a very recent work (https://arxiv.org/abs/2603.05630) introduces a new reconstruction-based metric, iFID, which is highly correlated with gFID. If large-scale experiments are not feasible, would it be possible to measure this metric instead, on datasets other than ImageNet?

---

> > > ### Author Response · Authors · 2026-04-06
> > >
> > > Thank you for bringing this to our attention. We would be happy to include this metric in our reconstruction results, I will be sure to run this experiment and update this conversation soon.

---

### Official Review · Reviewer_uUYo · 2026-03-12

**Soundness:** 2
**Presentation:** 3
**Significance:** 3
**Originality:** 2
**Overall Recommendation:** 4
**Confidence:** 4

**Summary:**

ViTok-v2 extends ViT-based autoencoders to native-resolution, ultra-large-scale, GAN-free stable training.
Through the combination of NaFlex, 2D RoPE, DINOv3 perceptual loss, and a 5B-parameter decoder, ViT-VAE surpasses CNN-VAE for the first time across high resolutions (512p-8K).

**Compliance With Llm Reviewing Policy:**

Affirmed.

**Final Justification:**

ViTok‑v2 explores the architectural design of a 4.5B‑parameter decoder, combining commonly used components such as NaFlex, 2D RoPE, and the DINOv3 perceptual loss. During inference, it adopts an existing sliding‑window attention mechanism to handle high‑resolution images. The ablation studies are relatively detailed and provide practical value.

The usefulness of an image tokenizer ultimately depends on whether the downstream generative model (e.g., DiT) is easy to train and whether it converges quickly. Strong reconstruction quality does not necessarily imply strong generation quality. In this regard, the experimental results are insufficient—there is a lack of visual comparisons and objective metrics against other models. This limits the practical significance of the work. The authors‘ rebuttal has largely addressed the concerns I raised.

**Key Questions For Authors:**

1. In Figure 2, subfigure (c) contains grid artifacts, but the textures and details are better restored. Subfigure (d) does not exhibit grid artifacts, but the overall result tends to be somewhat blurrier.
How should we interpret the fact that the 1024-token training results appear blurrier?

2. Flow-based models lack visualized result comparisons.

**Limitations:**

yes

**Strengths And Weaknesses:**

$\textbf{Strengths}$
1. ViTok‑v2 demonstrates that ViT‑VAE can surpass CNN‑VAE at high resolutions.
2. ViTok-v2 provides a training recipe that scales up to 5B parameters, which is highly valuable for industrial applications.

$\textbf{Weaknesses}$
1. Line 155-159, the radius r of the 2D sliding window attention (SWA) conflicts with the compression ratio r.
2. This work combines NaFlex, 2D RoPE, and DINOv3 perceptual loss, while the level of novelty is limited, its engineering value is high.

---

> ### Author Rebuttal · Authors · 2026-03-30
>
> We thank the reviewer for their thoughtful comments as well as recognizing the high engineering value and applicability of our work. Below, we provide responses to the reviewer's concerns.
>
> **On the notation conflict.** Thank you for catching this, we will disambiguate the SWA radius and compression ratio notation in the camera-ready version.
>
> **On novelty.** We highlight three specific novel contributions:
>
> (1) We are the first to use DINOv3 directly as a perceptual reconstruction loss for tokenizer training. This is not a drop-in replacement for LPIPS, as it requires a tile-based formulation using random 224x224 crops to handle DINOv3's fixed positional embeddings (Section 3, Appendix E). This enables stable GAN-free and LPIPS-free training at 5B parameters, where adversarial losses are known to be unstable. Finding 4 and Table 2 validate this.
>
> (2) No prior ViT tokenizer has shown native aspect-ratio-aware processing that generalizes to 8192p without fine-tuning. In this setting, our use of NaFlex with 2D RoPE is novel, and Table 5 shows strong reconstruction quality at resolutions where all CNN baselines run out of memory.
>
> (3) We present the first systematic study of how decoder capacity interacts with compression ratio (Figure 3), and how VAE scale interacts with flow model scale and channel count (Section 5.2, Figure 4). Our findings, such as the dependence of optimal channel count on generator capacity, are not derivable from prior work and directly inform future tokenizer design.
>
> **Clarifying differences between Figure 2's subfigures (c) and (d).** The perceived softening in (d) reflects the well-known perception-distortion trade-off.
> During the final 10% of training, the 1024-token NaFlex fine-tuning stage exposes the model to higher-resolution crops with a wider range of aspect ratios.
> This removes the grid artifacts visible in (c), but slightly changes the reconstruction behavior: pixel-level fidelity metrics such as PSNR and SSIM improve modestly, while the perceptual metric rFID increases slightly.
> We will add an explicit ablation table comparing 256-token NaFlex vs. 1024-token NaFlex at matched resolution in the camera-ready to quantify this trade-off precisely. Importantly, the grid artifacts in (c) are structurally incorrect and would propagate to downstream generation, making (d) the strictly preferred option for any practical visual generation pipeline.
>
> **On flow model visualizations.** We will add generated sample comparisons in the camera-ready. We note that generation quality is evaluated quantitatively via gFID and gFDD over 100k samples (Figures 4 and 5, Section 5), providing rigorous distributional comparisons.

---

> > ### Author Rebuttal · Reviewer_uUYo · 2026-04-02
> >
> > Thank you for the authors’ rebuttal. I still have a few remaining questions.
> >
> > **Regarding R1**:
> > If DINOv3 is used as a perceptual loss, it should be directly compared with other perceptual losses such as LPIPS or DISTS. LPIPS can also be used to train a tokenizer without adding a GAN loss.
> >
> > **Regarding R2**:
> > Many image VAEs can extrapolate to higher resolutions. The SD-VAE and DC-AE in Table 5 can also avoid OOM through tiling. ViTok-v2’s ability to support high resolutions similarly comes from using sliding-window attention, which is essentially another form of chunked attention computation.

---

> > > ### Author Response · Authors · 2026-04-06
> > >
> > > R1: Table 2 presents our comparison with LPIPS. Our results demonstrate that DINOv3 perceptual loss outperforms LPIPS and can further be used in combination with it for additional gains. We did not include a comparison with DISTS, as it is not commonly employed as a perceptual loss in modern VAE training pipelines.
> > >
> > > R2: We agree that CNN-based tokenizers can achieve this to some extent due to their inductive bias. However, we respectfully note that tiling is not mechanistically equivalent to sliding window attention, as tiling requires predefined boundaries for the tokenizer, and these boundaries do not produce seamless transitions in the way SWA does. That said, we appreciate the suggestion and will include additional tiling results from CNN-based approaches to better illustrate this distinction and present it as a potential complementary solution. Thank you for raising this point.

---

### Official Review · Reviewer_ft8Q · 2026-03-13

**Soundness:** 3
**Presentation:** 3
**Significance:** 3
**Originality:** 2
**Overall Recommendation:** 4
**Confidence:** 4

**Summary:**

This paper considers the concept of scaling ViT–based visual tokenizers to very large parameter counts and enabling flexible-resolution processing. It also introduces DINOv3-based perceptual reconstruction loss replacing LPIPS and adversarial training. The paper presents extensive experiments on reconstruction benchmarks and downstream flow-based generations. The results suggest that decoder scaling significantly improves reconstruction quality and that larger generators can better exploit high-channel latent representations.

**Compliance With Llm Reviewing Policy:**

Affirmed.

**Final Justification:**

Thanks for the rebuttal. I remain my positive rating.

**Key Questions For Authors:**

Please refer to the Weakness I list.

**Strengths And Weaknesses:**

**Strengths**

1. This paper conducts a large-scale empirical study. The tokenizer scaling is interesting and carefully evaluated across multiple compression ratios and model sizes.
2. Experiments are comprehensive with a wide range of evidence: (1) encoder/decoder architecture ablations (2) loss function comparisons (3) resolution generalization experiments (4) joint scaling experiments with downstream flow models

**Weaknesses**

1. Although this paper provides practical insights with solid experiments, many components of the proposed method appear to be combinations of existing techniques rather than fundamentally new ideas: (1) NaFlex training originates from NaViT-style architectures (2) 2D RoPE positional embeddings are widely used in recent transformer models (3) perceptual losses based on pretrained representation models are well established.
2. In Section 4.3, the author only states the conclusion that “models transfer zero-shot to SWA,” without providing any explanation or experimental evidence.

Overall, I believe this paper provides solid evidence for scaling ViT-based tokenizers and represents a meaningful step toward their practical use.

---

> ### Author Rebuttal · Authors · 2026-03-30
>
> We thank the reviewer for their comments and address their concerns below.
>
> **On novelty.** We respectfully note that while individual components come from existing work, our specific contributions are novel:
>
> (1) We are the first to use DINOv3 directly as a perceptual reconstruction loss for tokenizer training. This is not a drop-in replacement for LPIPS, as it requires a tile-based formulation using random 224x224 crops to handle DINOv3's fixed positional embeddings (Section 3, Appendix E). This enables stable GAN-free and LPIPS-free training at 5B parameters, where adversarial losses are known to be unstable. Finding 4 and Table 2 validate this.
>
> (2) No prior ViT tokenizer has shown native aspect-ratio-aware processing that generalizes to 8192p without fine-tuning. In this setting, our use of NaFlex with 2D RoPE is novel, and Table 5 shows strong reconstruction quality at resolutions where all CNN baselines run out of memory.
>
> (3) We present the first systematic study of how decoder capacity interacts with compression ratio (Figure 3), and how VAE scale interacts with flow model scale and channel count (Section 5.2, Figure 4). Our findings, such as the dependence of optimal channel count on generator capacity, are not derivable from prior work and directly inform future tokenizer design.
>
> **On zero-shot SWA transfer.** Table 5 provides direct experimental evidence of effective zero-shot SWA transfer: ViTok-v2 trained with full attention achieves virtually identical reconstruction metrics when using SWA at inference across 1024p--8192p resolutions. We will add a brief explanation of the underlying mechanism (2D RoPE preserving relative position information within local windows) in the revision.

---

> > ### Author Rebuttal · Reviewer_ft8Q · 2026-04-03
> >
> > Thanks for the rebuttal. I remain my positive rating.

---

### Official Review · Reviewer_EsSg · 2026-03-13

**Soundness:** 3
**Presentation:** 3
**Significance:** 3
**Originality:** 3
**Overall Recommendation:** 5
**Confidence:** 3

**Summary:**

This paper presents ViTok-v2, a Vision Transformer-based autoencoder for compressing images into latent tokens that are used by generative models. The proposed model uses an asymmetric architecture with a shallow encoder and a large transformer decoder scaled up to 5B parameters. Experiments show that decoder scaling improves reconstruction quality, and  the tokenizer achieves improved reconstruction performance at 512p and higher resolutions.

**Compliance With Llm Reviewing Policy:**

Affirmed.

**Key Questions For Authors:**

Do the authors have further insights or analysis explaining why the decoder capacity dominates encoder capacity in determining reconstruction and generation quality?

**Limitations:**

Yes

**Strengths And Weaknesses:**

Strengths:
+ The paper reports elaborated empirical validation for the proposed model. It also includes ablations that quantify the effects of all components, i.e. decoder scaling, encoder depth, loss design, and resolution training strategies.
+ The paper is well written, clear, and well organized. It provides  clear and detailed description of the model architecture, training procedures, hyperparameters, and loss functions in the appendix, which improves transparency and reproducibility.
+ The use of NaFlex variable-resolution training and sliding window attention is reasonable.

Weaknesses:
- Many components of the work build heavily on existing ideas, and the contribution mainly is limited to scaling and combining these techniques.
- The paper doesn't provide enough evidence on why larger decoders improve downstream generation even when reconstruction metrics are similar.
- The discussion of related work could benefit from clearly differentiating the work from closely related tokenizer methods.

---

> ### Author Rebuttal · Authors · 2026-03-30
>
> We thank the reviewer for their thoughtful and constructive feedback, and for recognizing our thorough empirical validation and clear presentation. We address each concern below.
>
> **On novelty beyond scaling and combining existing techniques.** We respectfully highlight several contributions that go beyond combining prior work:
>
> (1) To our knowledge, we are the first to use DINOv3 directly as a perceptual reconstruction loss for training visual tokenizers. This replaces both LPIPS and adversarial GAN objectives, enabling stable training at billion-parameter scale which is a qualitatively different training recipe, not merely a substitution (Section 4.4, Table 2).
>
> (2) We scale ViT image tokenizers to 5B parameters (14x larger than prior work) and reveal a previously uncharacterized interaction: decoder scaling benefits *increase* with compression ratio, with implications for how tokenizers should be designed at different operating points (Figure 3). A scale like that has not been tried before in the context of visual compression and reconstruction.
>
> (3) We conduct the first joint VAE--flow model scaling study, demonstrating that optimal channel count depends on generator capacity. This is a finding not established in prior work (Section 5.2, Figure 4). We also demonstrate zero-shot full-attention to SWA transfer, enabling 8K reconstruction where all CNN baselines OOM (Table 5).
>
> **On why larger decoders improve downstream generation.** We respectfully disagree that insufficient evidence is provided and would like to point out how our paper defends this point thoroughly across multiple experiments.
>
> For reconstruction, Figure 3 shows that scaling the decoder from B (88M) to T (4.5B) consistently improves PSNR, SSIM, rFID, and rFDD across all compression ratios. Crucially, the dotted lines in Figure 3 show that doubling encoder depth (d=4 vs d=8) yields negligible improvement, showing that decoder capacity, not encoder capacity, is the limiting factor. This is further supported by Table 1 (Section 4.1), where reducing encoder depth from 4 blocks to just 1 block has minimal SSIM impact.
>
> For generation, Figure 5 directly addresses how larger decoders improve downstream generation in relation to reconstruction performance. It plots reconstruction quality against generation quality for both the 5B VAE (red) and 350M VAE (blue) across 16/32/64 channels. The 5B curves consistently sit in the lower-left region of the plot, achieving both better reconstruction and better generation. More importantly, the 5B VAE achieves 1--2 gFID improvement over the 350M even at comparable reconstruction metrics, demonstrating that larger decoders benefit generation through mechanisms beyond reconstruction fidelity alone. Section 5.3 discusses this finding explicitly.
>
> Taken together, Figures 3 and 5 provide comprehensive evidence from both the reconstruction and generation perspectives that decoder scaling has a large effect in improving downstream reconstruction and generation quality.
>
> **On related work differentiation.** We agree the related work section would benefit from sharper differentiation between similar tokenizer methods. We will revise our writing so that we more explicitly contrast our approach with closely related tokenizers across loss design, regularization strategy, resolution handling, and scale.

---

> > ### Author Rebuttal · Reviewer_EsSg · 2026-04-04
> >
> > The authors have addressed my main concerns by providing clearer explanations of the method and enhancing the overall presentation. Their effort to resolve previously unclear points is appreciated.

---

### Decision · Program_Chairs · 2026-04-30

**Decision:**

Accept (regular)

**Comment:**

This work presents ViTok-v2, a Vision Transformer (ViT) based tokenizer that improves upon the prior work ViTok. The key technical ingredients include native resolution support (NaFlex and 2D RoPE) and the replacement of VGG-based perceptual loss with DINOv3 (with the removal of GAN loss), The resulting tokenizer is further scaled up to 5B parameters and evaluated on high-resolution cases (8K).

While all reviewers appreciate the careful and detailed empirical experiments, they share a common concern that most of the ingredients already exist in the literature, compromising the novelty of ViTok-v2. Additionally, some reviewers were initially uncertain about specific method and experiment details (e.g., notations and figures). The provided rebuttal effectively assuaged these concerns to some degree.

In the end, two reviewers remained concerned about the insufficient experiments regarding image generation, a critical downstream application for tokenizers. After checking the submission, reviewer comments, and author rebuttals, the AC shares the same concern that ViTok-v2 has not yet demonstrated sufficient effectiveness for image generation, making the necessity of scaling up the tokenizer parameters to 5B questionable. Nevertheless, the AC appreciates the engineering improvements and rigorous experiments presented by the submission. As a result, the AC recommends Weak Accept and encourages the authors to incorporate the reviewer comments (particularly, further exploration of generation tasks) to improve their final work.